# Consumers' Attitudes towards Online Advertising: A Model of Personalization, Informativeness, Privacy Concern and Flow Experience

**Li Mo [1], Xiaosan Zhang [2], Yabin Lin [1], Zhenghui Yuan [2,*] and Zengjun Peng [3]**

[1] School of Communication, Fujian Normal University, Fuzhou 350117, China
[2] Research Center for Belt and Road Financial and Economic Development, Xiamen National Accounting Institute, Xiamen 361000, China
[3] Department of Mass Communications, St. Cloud State University, St. Cloud, MN 56301, USA
[*] Correspondence: yzh@xnai.edu.cn

**Abstract:** Online personalized advertising has been widely adopted in China in the recent years, leading to both positive and negative results. This study endeavors to examine the impact of perceived personalization of online advertising on consumers' attitudes toward advertising. A total of 472 questionnaires were administered and analyzed using Structural Equation Modeling. The results show that perceived personalization exerts a positive impact through perceived informativeness, and a negative impact through privacy concerns. However, the positive effect was determined to be stronger and thus perceived personalization had an overall positive impact. Additionally, the results showed that both effects were mediated by flow experience. The practical and theoretical implications of the findings were discussed.

**Keywords:** online advertising; personalization; informativeness; privacy concern; flow experience

## 1. Introduction

With the prevalence of social media and online communication, the media environment has become increasingly fragmented. Chinese advertisers, just like advertisers in many other countries, have adopted algorithms and mathematical methods as means to engage website users and provide them with targeted information. The widespread utilization of these methods is reflected in the investment made in the online advertising industry in China. According to a marketing research report, 1110.1 billion RMB were invested in online advertising in 2022 with an 18.8% growth rate [1]. Online advertising, which is generated based on website users' data such as socio-demographics, interests tags, location, brand communities, healthy situation recorded by fitness trackers, as well as behavior data such as browsing history, keywords searched, online shopping records, talking contents detected by surveillance-enabling devices, is prevalent on various Chinses social network sites (SNSs), including Weibo, WeChat and DouYin, as well as e-commerce platforms, including Taobao, Jingdong and Suning. In this context, it is imperative to investigate the potential effects of online advertising on the behaviors of Chinese consumers.

Online advertising is highly personalized. The data used in the personalization process is sometimes collected with explicit consent, which means people volunteer their information for a better online experience such as more relevant options, greater convenience, and access to free content [2]. Indeed, a survey conducted in China determined that 46% of the participants hold a positive attitude toward online ads, and more than 60% of the consumers showed varying degrees of willingness to have their online activity tracked [3]. However, personal data can also be obtained from third parties or shared among various websites without people's consent [4]. A survey revealed that half of the users interviewed held the belief that their personal data were extensively collected and

utilized improperly [5]. This belief is a manifestation of concerns regarding the potential infringement of the right to protect the confidentiality of personal information, commonly referred to as privacy concerns [6]. Therefore, although online advertising can generate more favorable consumer responses as it is customized to align with consumers' interests and preferences [7], it also gives rise to negative outcomes [6–8]. It is evidenced by the results of these surveys and studies that both perceived informativeness and privacy concerns play an important role in consumer attitude to advertisement. However, the exact impact of these factors, particularly as they are both influenced by personalization, remains unclear. This study aims to examine the effect of perceived personalization on consumers' attitudes toward online advertising by evaluating both the favorable aspect (perceived informativeness) and the unfavorable aspect (privacy concern). The objective of this study is to shed light on the relative impact of these two factors, offering an insight into whether consumers' attitudes would be more likely to be influenced by the benefit or the threat.

This study also attempts to uncover the role of flow experience. Flow experience, characterized by perceived control, curiosity, intrinsic interest, interactivity, and focused attention, is defined as the holistic sensation that individuals experience when they engage in an activity with total involvement [9,10]. People can easily experience flow when browsing or viewing the web. As a part of the web, online advertising can provide consumers with messages meeting their preferences, and allow them to decide whether to engage with such advertisements or not. This means sophisticated advertising features are designed to foster user engagement and control, thus augmenting their flow experience. On the other hand, the well-customized nature of online advertising can also result in a weakened flow experience as it may evoke feelings of intrusiveness, causing users to become concerned about their privacy and feel distracted. Given that flow was determined to be one of the key factors influencing people's attitudes and behavior in the online context [11,12], it is crucial to explore the impact of a personalized process on flow, and subsequently on consumers' attitudes.

To sum up, the objective of this study is to formulate and analyze a comprehensive model for personalized online advertising. Through this examination, the present research makes a contribution to the existing literature on online advertising by providing new insights in two distinct ways.

First, this study endeavors to increase the understanding of the internal mechanisms through which personalized online advertising influences consumers. The investigation specifically focuses on examining both perceived informativeness as a positive internal response and privacy concerns as a negative internal response concurrently, as opposed to most previous studies which have primarily examined either one in isolation or under certain contexts [13]. We then establish a connection between these internal responses and consumers' attitudes toward the ads. Through this examination, our work aims to clarify the inconsistent results that exist in the literature regarding the effects of online advertising.

Second, previous research has adopted the notion of privacy calculus to comprehend the manner in which individuals evaluate the benefits and risks associated with online advertising [14]. These studies indicate that when people perceive more benefits than costs, they are more likely to exhibit favorable attitudes toward the advertisement and the associated brand [15]. However, web users may not have the time or the ability to conduct the rational calculation in reality. By introducing flow experience into the model, this study might provide a more realistic representation of the underlying process.

From a practical perspective, the research holds significant implications for businesses and organizations seeking to target and influence Chinese consumers through online advertising. As online advertising keeps on growing in China at a fast pace, it is crucial for marketers to comprehend consumer perceptions and reactions toward online advertising. This study provides specific recommendations for advertising practitioners, highlighting the importance of providing consumers with informative messages that are based on the data while avoiding actions that trigger privacy concerns. Additionally, it is recommended that efforts should be made to create opportunities for consumers to experience flow.

## 2. Literature Review and Hypotheses

### 2.1. The Benefits of Online Advertising: Perceived Personalization and Informativeness

Online advertising, which is also called online behavioral advertising (OBA), refers to "the practice of monitoring people's online behavior and using the collected information to show people individually targeted advertisement" [16]. Fundamentally, the idea behind this concept is to treat each message recipient as unique and tailor the message based on the recipient's characteristics [17]. A vast array of data such as demographic information, interests, online activity, media consumption, and communication content, among others, can be utilized to determine the content of the advertisement and the intended recipient [18,19]. Therefore, online advertising is believed to be tailored based on the customers' needs and is highly personalized. It should be noted that the word "personalized", though seems as objective as what should be regarded as personalized, has been defined by law and regulation, and can vary because people perceive some types of data as more sensitive [20]. Therefore, perceived personalization, rather than the actual level of personalization, is crucial in shaping individuals' views and responds to online advertising. In recent years, there has been a growing interest in personalization within the realm of online advertising [16]. Studies have demonstrated that personalization could increase ad effectiveness such as improved brand engagement, higher click-through rates and favorable attitudes toward the advertisement or brand, and a rise in purchase intentions [16,20,21]. One of the factors that has been explored concerning personalization is informativeness, which encompasses the consumers' perception that the content of an advertisement is informative regarding the product/service being advertised [22]. When customers feel that an advertising message is relevant to them, they are more likely to consider it useful [23]. Given that online ads are tailored based on consumers' data, for example, based on information from social media [20], it is anticipated that they will provide consumers with the information they seek and allow them to quickly focus on their desired information [24]. As a result, we expect that when a consumer is confronted with online advertising, the more personalized the advertisements feel, the more informativeness they present.

The primary purpose of advertising for customers is to gather information regarding products or services. According to the advertising value model [25], informativeness was one of the important factors affecting consumers' perceptions of the relative value or utility of advertising. Generally speaking, customers anticipated advertising to be more informative than mere product placement [26]. When people perceived ads as supplying timely, relevant information, or having been told about products when they needed the information, they would view the ads as valuable [25]. A positive relationship between informativeness and consumers' attitudes toward ads had been determined in specific contexts such as mobile marketing [12,27] and Facebook [13]. Therefore, we expect such a positive relationship to exist in a more general online context. Based on these findings, it is hypothesized that:

**H1.** *Perceived personalization positively affects consumers' perception of ad informativeness.*

**H2.** *Consumers' perception of ad informativeness positively affects consumers' ad attitudes.*

### 2.2. The Disadvantage of Online Advertising: Perceived Personalization and Privacy Concern

The utilization of personalized advertising can also lead to the feeling of threat to freedom, causing negative responses, such as irritation, negative cognitions, or avoidance [8]. Research has indicated that privacy concerns play a critical role in the efficacy of personalized advertising [28]. Privacy concerns pertain to the extent to which consumers are worried about the unauthorized dissemination of their personal information [7]. With the development of internet technology, more than 1 billion people in China have access to the online space, accounting for 71.6% of the population [5]. Due to the increasing frequency of data hacking and leakage incidents, consumer privacy has become a growing concern. Previous studies have established that privacy concerns have a negative impact

on consumers' attitudes toward online advertising [27]. People with a high level of privacy concerns might feel anxious and insecure when viewing online ads, leading to a decrease in their willingness to receive and use such ads. Such a conclusion has been drawn from both western and eastern countries [27,29] and can be attributed to psychological reactance theory, which posits that consumers would perceive their privacy as threatened when they viewed online advertising as a way to direct their life by tracking their online behavior [7]. This, in turn, may trigger reactance and motivate them to modify their attitudes to restore freedom [30]. It could also be explained from the aspect of the dual-process model of approach and avoidance motivation, which suggests individuals under a behavioral inhibition system would have an aversion motivation to avoid negative stimuli [31], which is the loss of privacy in this case. Privacy concerns are contingent on context, with previous studies demonstrating that highly personalized ads tend to elicit high levels of privacy concerns among consumers [15,32]. Consequently, we expect that when consumers view ads that align with their interests and needs, their privacy concerns are more likely to be activated.

**H3.** *Perceived Personalization positively affects consumers' privacy concerns.*

**H4.** *Consumers' privacy concerns negatively affect consumers' ad attitudes.*

### 2.3. The Role of Flow Experience

Flow is "the holistic sensation that people feel when they act with total involvement" [9]. It was studied in many activities in real life such as sports [33], reading [34], and shopping [35], then extended to hypermedia computer-mediated environment, where humans interacted with computers and the Web [36,37]. Research has shown that individuals who experience flow online become highly concentrated and "the state of mind arising as a result is extremely gratifying" [37]. Several studies that apply flow theory to online consumption have discovered that flow experience can positively influence marketing-related online behaviors, such as revisiting the web, web and brand attitudes [38], online purchasing [11], and purchase intention [39]. The interactive nature of the internet is believed to play a role in promoting flow experiences among users [40]. As Hoffman and Novak [36] suggested, the success of online marketers depends on their ability to create opportunities for consumers to experience flow. One antecedent of flow identified in the literature is value-added search [41]: once consumers viewed the personalized ads and processed the message as something worthwhile, they might feel present in the virtual environment, focusing on the contents of ads and disregarding irrelevant thoughts [27]. This can result in a state of flow. As such, we posit that consumers exhibit a stronger positive attitude toward advertising when they experience flow, and they are more likely to experience flow when receiving informative ads. However, personalized advertising also raises consumer privacy concerns [7], which can lead to feelings of intrusiveness and worries about personal data misuse [42], potentially distracting consumers and undermining their ability to focus, leading to weak flow experiences and negative attitudes toward advertisements. Therefore, it is hypothesized that

**H5.** *Perceived ad informativeness positively affects consumers' flow experience.*

**H6.** *Privacy concerns negatively affect consumers' flow experience.*

**H7.** *Consumers' flow experience positively affects consumers' ads attitude.*

In summary, we postulate that the perceived personalization of online ads has a positive impact on ad attitudes by augmenting the flow experience via the perception of informativeness, whereas it has a negative impact by diminishing the flow experience due to privacy concern. A hypothesized model is illustrated in Figure 1.

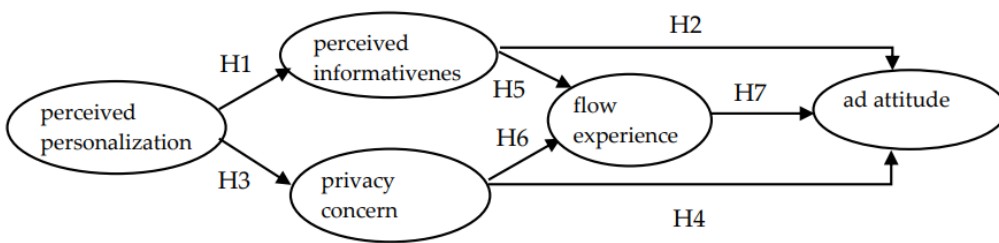

**Figure 1.** Hypothesized Model.

## 3. Research Methodology

### 3.1. Questionnaire Design

In this study, primary data were collected through the utilization of a structured questionnaire. The questionnaire measured key variables in the study, including perceived personalizing, perceived advertising informativeness, flow experience, and privacy concern. The measurement of these constructs was carried out through the use of five-point Likert scales, where each item was rated on a scale ranging from "strongly disagree" (1) to "strongly agree" (5). The questionnaire also included demographic questions relating to age and gender.

### 3.2. Data Collection

An investigation was carried out to test the propositions put forth in this research. The questionnaire was uploaded to the web, to be divulged online. The useable data were mainly collected from college students from different cities in China, through the method of convenient sampling. While the selection criteria for participants may pose a potential constraint to the generalization of the results, it should be noted that college students in China were born in an era of the internet and had enough experience viewing personalized ads. In addition, college students are becoming an important market for targeting marketers [7]. Furthermore, the use of a student sample could help reduce error variance due to its homogeneity, resulting in a stronger test of theory [43]. A total number of 622 was distributed and 536 questionnaires were collected. The responses to each question were carefully scrutinized. The questionnaire was regarded as invalid if any of the following conditions was met: (1) either answer for the two discriminative questions "Are you a college student now?" and "Have you seen any online adverting in the recent month?" is yes; (2) the answering time is shorter than 2 min; (3) the answer shows an obvious pattern. The final sample comprised 472 individuals, which could be considered adequate [44]. A total of 33.5% (158) of the respondents were male and 66.5% (314) were female. A total of 97.7% (461) were 18–25, which is the typical age for college students. A total of 0.8% (4) were under the age of 18, and 1.5% (7) were 25–30 years old.

### 3.3. Measure

The measures for perceived personalization were adapted with four items modified from Kim and Ham [12]. Advertising informativeness was measured with four items modified from Ducoffe [25] and Kim and Ham [12]. Flow experience was measured by a scale of four items revised from Novak [37] and Martins [27]. Privacy concern was measured with a scale of five items from Baek [7]. Advertising attitude was measured by four items adopted from Simons and Carey [45]. As all the measurements were adopted from studies written in English, a pilot survey consisting of 50 participants was conducted to refine the questionnaire written in Chinese. Minor modifications were made to the final questionnaire (see Table 1 for items). All the scales were reliable as Cronbach's alpha values exceed the reference value of 0.7 (Cronbach's $\alpha$ for perceived personalization = 0.850, Cronbach's $\alpha$ for informativeness = 0.900, Cronbach's $\alpha$ for flow experience = 0.900, Cronbach's $\alpha$ for privacy concern = 0.960, Cronbach's $\alpha$ for advertising attitude = 0.950).

**Table 1.** Convergent validity.

| Construct | Factor Loading | CR | AVE |
|---|---|---|---|
| Perceived Personalization | | | |
| I feel online advertisements are tailored to me (PP1) | 0.825 | | |
| I feel online advertisements are customized to my needs (PP2) | 0.802 | 0.833 | 0.563 |
| I feel online advertisements are personalized according to my profile (PP3) | 0.603 | | |
| I feel online advertisements are delivered in a timely way (PP4) | 0.740 | | |
| Advertising informativeness | | | |
| Online advertising provides timely information on products or services (Inf1) | 0.855 | | |
| Online advertising supplies relevant information on products or services (Inf2) | 0.827 | 0.900 | 0.694 |
| Online advertising is a good source of information (Inf3) | 0.816 | | |
| Online advertising is a good source of up-to-date products or services information (Inf4) | 0.816 | | |
| Flow experience | | | |
| I completely concentrate on online advertising while I look at it (Flo1) | 0.796 | | |
| While I read online advertising, time seems to pass by very quickly (Flo2) | 0.756 | 0.883 | 0.625 |
| While I watch online advertising, nothing seems to matter (Flo3) | 0.819 | | |
| While I view online advertising, I feel totally captivated (Flo4) | 0.862 | | |
| Privacy concern | | | |
| When I receive online adverting, I feel uncomfortable when information is shared without permission (PC1) | 0.840 | | |
| I am concerned about the misuse of personal information (PC2) | 0.905 | 0.933 | 0.764 |
| It bothers me to receive too much advertising material of no interest (PC3) | 0.876 | | |
| I feel fear that information may not be safe while stored (PC4) | 0.920 | | |
| I think companies share information without permission (PC5) | 0.737 | | |
| Advertising Attitude | | | |
| I think online advertising is positive (AD1) | 0.900 | | |
| I think online advertising is good (AD2) | 0.901 | 0.939 | 0.803 |
| I like online advertising (AD3) | 0.887 | | |
| I think online advertising is desirable (AD4) | 0.874 | | |

## 4. Analysis of Results

### 4.1. Descriptive Statistics for Key Variables

The mean score for privacy concern (M = 4.11) indicated that the survey participants were highly concerned about privacy. Survey participants view online advertising as tailored and customized to their needs to a high extent (M = 3.98), thus perceiving it as informative (M = 3.15). Their flow experience (M = 2.46) and attitudes to online advertising (M = 2.61) were lower than the mid-point of the 5-point scale.

### 4.2. Common Method Bias Test

In this study, common method bias may have been introduced due to the fact that most of the data were collected at one point in time. To mitigate this issue, a Harman single-factor test was conducted. Results indicate that the explained variance percentage of the single factor is 36.28%, which is less than 50%. To further address this potential issue, a latent common method factor was added to the hypothesized six-factor model. This model fitted the following data: $\chi^2(178) = 412.176$, $\chi^2/df = 2.316$, CFI = 0.968, TLI = 0.963, IFI = 0.968, AGFI = 0.901 and RMSEA = 0.053, almost same with the hypothesized five-factor model. Therefore, the common method bias in this study does not affect the results.

### 4.3. Assessment of the Measurement Model

The measurement model was analyzed using structural equation modeling (SEM) by IBM-SPSS-AMOS version 18.0, https://www.ibm.com/support/pages/downloading-ibm-spss-modeler-180 (accessed on 7 January 2023). Maximum Likelihood Estimation was utilized to evaluate whether the measurement items for each construct were correctly loaded onto their respective constructs. Fit indices for the models were interpreted according to the criteria suggested by Garson [46]. Overall goodness-of-fit indices were satisfactory

since they are far above the conventional threshold [47]: 2.303 $\chi^2/df$, 0.053 Root Mean Square Error of Approximation (RMSEA), 0.923 Goodness of Fit Index (GFI), 0.901 Average Goodness of Fit Index (AGFI), 0.968 Confirmatory Fit Index (CFI), 0.969 Incremental Fit Index (IFI), 0.963 Tucker-Lewis Coefficient (TLI) (see Figure 2).

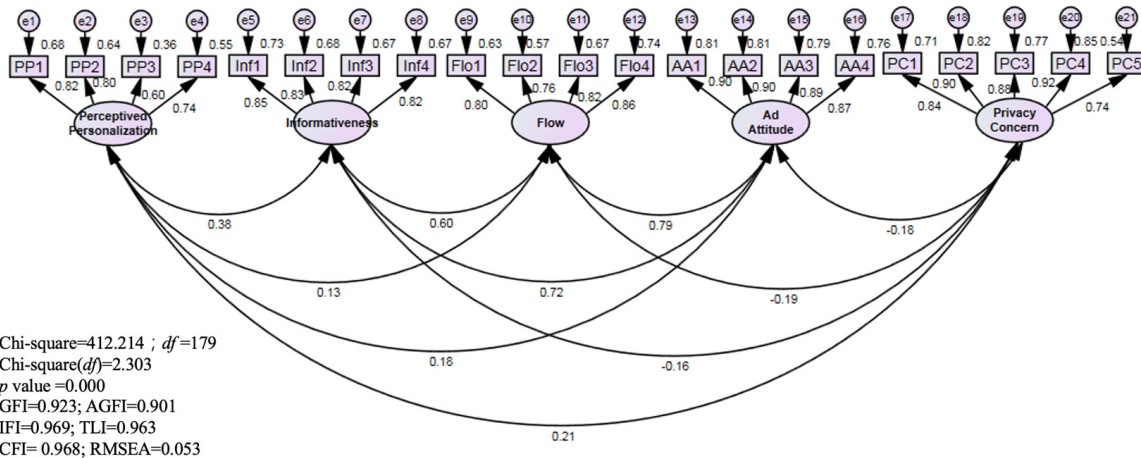

**Figure 2.** Measurement model and standard estimates.

The convergent validity was evaluated by three criteria: average variance extracted (AVE) for each construct, composite reliability (CR) of measurement items, and factor loading of measurement items. The AVE value for each construct ranged from 0.563 to 0.803, which was greater than the benchmark of 0.5. Composite reliability for each construct exceeded the threshold of 0.7, ranging from 0.833 to 0.956. Standardized factor loading coefficients for all measurement items were greater than the recommended level of 0.5. All standardized item loadings were significant at 0.001 [48].

The discriminant validity was assessed with the standard that the AVE of each construct exceeds the square of the standardized correlations between the two constructs [47]. No discriminant validity issues were detected. Thus, both convergent validity and discriminant validity were established (see Tables 1 and 2).

**Table 2.** Discriminant validity.

| Construct | M | SD | (1) | (2) | (3) | (4) | (5) |
|---|---|---|---|---|---|---|---|
| Perceived personalization | 3.98 | 0.81 | 0.750 | | | | |
| Advertising informativeness | 3.15 | 0.98 | 0.378 ** | 0.949 | | | |
| Flow experience | 2.46 | 0.98 | 0.128 ** | 0.599 ** | 0.949 | | |
| Privacy concern | 4.08 | 0.95 | 0.211 ** | −0.157 * | −0.192 * | 0.979 | |
| Advertising attitude | 2.61 | 1.03 | 0.179 ** | 0.725 ** | 0.790 ** | −0.185 ** | 0.975 |

$* p \leq 0.05$, $** p \leq 0.01$.

*4.4. Assessment of Structural Model and Hypothesis Testing*

IBM-SPSS-AMOS version 18.0 was employed to assess the structural model. The structural model fit was acceptable with AGFI slightly below the 0.9 benchmark: 2.434 $\chi^2/df$, 0.055RMSEA, 0.918 GFI, 0.959 TLI, 0.896 AGFI, 0.965 CFI. The results of the hypotheses testing are illustrated in Figure 3.

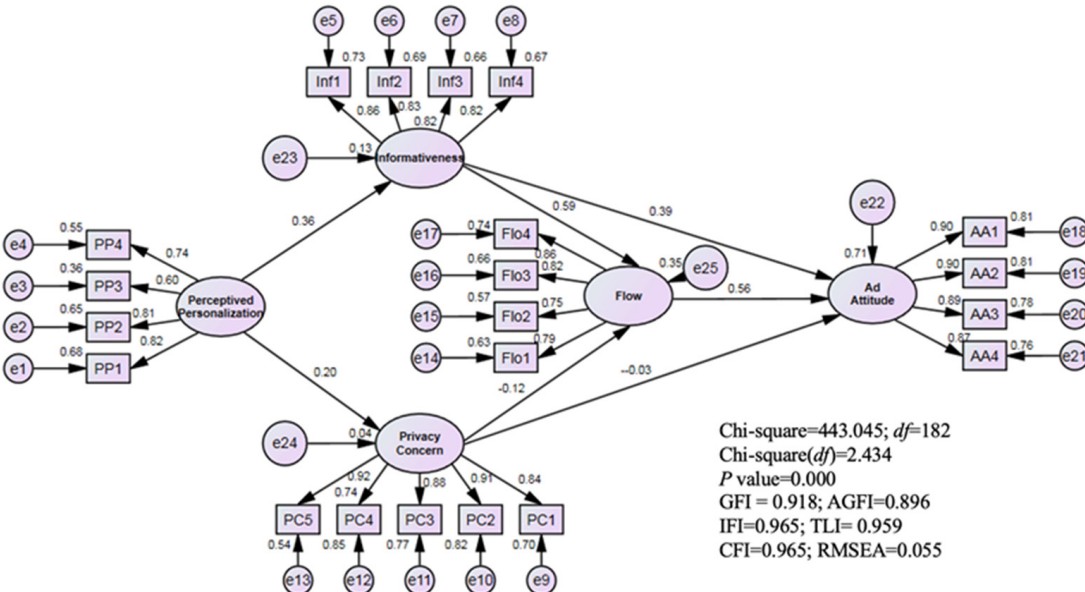

**Figure 3.** Structural Model Test Result.

First, perceived personalization positively and significantly affects perceived ad informativeness (β = 0.361, *p* < 0.001), and perceived ad informativeness significantly affects ad attitude (β = 0.391, *p* < 0.001). Thus, H1 and H2 are supported. This result indicates that ad informativeness plays a mediation role, as consumers who perceive an advertisement as more personalized perceive it as more informative, and have a stronger positive attitude towards advertising.

Second, perceived personalization positively affects privacy concerns (β = 0.197, *p* < 0.001), and H3 is supported. Privacy concerns have no significant effect on ad attitude (β = −0.28, *p* > 0.1). H4 is not supported.

Third, perceived ad informativeness positively affects consumers' flow experience (β = 0.587, *p* < 0.001), and flow experience positively affects consumers' ad attitude (β = 0.555, *p* < 0.001). Thus, H5 and H7 are supported. These results indicated that flow experience played a mediation role between perceived informativeness and ad attitude.

Fourth, privacy concerns significantly and negatively affect consumers' flow experience (β = −0.119, *p* < 0.001), and flow experience positively affects consumers' ads attitude (β = 0.555, *p* < 0.001). Thus, H6 is supported. These results indicate that the flow experience plays a mediation role between privacy concerns and ad attitude. The greater the extent of privacy concerns activated in the consumer, the lesser the flow in the consumers' experience, and the lesser the positive ad attitude.

### 4.5. Test of Mediation Effect Using Bootstrapping

Following Cheung and Lau's procedure [49], we conducted a bootstrapping analysis with SPSS to test the mediation effect. To be specific, we tested the significance of indirect effects by examining the 95% confidence intervals using Preacher and Hayes' PROCESS macro model 6 [50]. In this study, we requested 5000 bootstrap samples with replacements drawn from the original data set. As shown in Table 3, all four indirect paths were supported (*p* < 0.01). The direct paths from perceived personalization to ad attitude were insignificant (*p* > 0.05). The findings provided evidence for indirect-only mediation. It should be noted that the indirect effect of perceived advertising personalization on ad attitude through ad informativeness is stronger than that through privacy concerns.

**Table 3.** Testing for Mediation with Bootstrapping.

| Path | Estimate | SE | *p*-Value | Lower (BC 95% CI) | Upper (BC 95% CI) |
|---|---|---|---|---|---|
| PP-Inf-AA | 0.15 | 0.029 | 0.000 | 0.097 | 0.210 |
| PP-Inf-Flo-AA | 0.106 | 0.214 | 0.000 | 0.067 | 0.151 |
| PP-PC-AA | −0.022 | 0.013 | 0.010 | −0.051 | −0.002 |
| PP-PC-Flo-AA | −0.036 | 0.014 | 0.000 | −0.067 | −0.013 |

Note: BC = bias corrected; CI = confidence interval; SE = standard error; PP= perceived personalization; PC = privacy concerns; AA = advertising attitude; Inf =perceived informativeness; Flo = flow.

## 5. General Discussion

### 5.1. Discussion of the Results

This study aimed to explore the effect of online personalized advertising in the rapidly growing Chinese market, where the prevalence of online personalized advertising has increased in recent years. Despite this trend, there has been a scarcity of research conducted in this area and contradictory results concerning the effects of online advertising were detected. Viewing personalization as the foundation of online advertising, this study sought to explore its impact. In line with previous research in western contexts [20,21], the results indicated that perceived personalization positively affected consumers' attitudes toward online advertising in the rapidly growing Chinese eastern market. Notably, the ads in our study were abstract, without clarifying the type or degree of personalization. It was therefore interesting to observe that a general form of personalization can elicit positive effects when it is perceived as personalized by consumers. This finding is consistent with previous findings [6], which suggests that a minimal degree of personalization or very general personalization based on gender can induce a positive attitude.

As to the effect of perceived personalization on ad attitude, this study constructed and examined a model highlighting perceived informativeness, privacy concern, and flow experience. Overall, the study confirmed most of the hypotheses and revealed four mediation paths. The "personalization–privacy paradox" which has been established as simultaneously increasing and decreasing consumer interaction with marketers [18] was observed in our study. To be specific, web users are more inclined to perceive tailored advertisements as providing information that is relevant to their interests and needs, thus eliciting greater engagement and favorable attitudes towards the ads (as suggested by H1, H2, H5, H7). Meanwhile, the utilization of personalized information in such ads also induces feelings of distraction, as users may experience concerns about the potential misuse of their personal data (as suggested by H3, H6, H7). As the positive effect is stronger (0.256) than the negative one (0.058), an overall positive attitude was determined. The findings help us understand the contradictory findings concerning effects of personalized advertisements.

The findings of this study support the importance of flow experience, as indicated by the significant effects of H5, H6, and H7. Specifically, flow experience was determined to fully mediate the impact of ad informativeness on ad attitude, while also partially mediating the influence of privacy concerns. These results indicate that consumers who perceive themselves as being captivated by online advertisements and exhibit a sense of concentration are more inclined to develop a favorable attitude towards such advertisements, even in the face of concerns about privacy.

### 5.2. Theoretical and Practical Implications

From a theoretical standpoint, this research study contributes to the existing literature on the dual-process model of approach and avoidance motivation by demonstrating that consumers in Eastern cultures display both positive (approach) and negative (inhibition) attitudes towards personalized advertisements. The study findings further indicate that the positive impact of perceived personalization through perceived ad informativeness outweighs the negative impact of privacy concerns, resulting in an overall positive attitude towards personalized ads.

Moreover, this study extends the processing mechanism of personalized advertising by revealing that the flow experience plays a crucial role in influencing consumers' attitudes towards advertisements. Specifically, due to insufficient time or cognitive resources, consumers may not have the capacity to engage in rational calculations. Instead, the flow experience, which is amplified by informativeness but weakened by privacy concerns, places consumers in an online state that shapes their attitudes towards online advertising. The incorporation of the flow experience into the personalized advertising model offers an enhanced approach to explicate the effects of online advertising based on the privacy calculus theory, thus rendering it more verisimilar to real-life situations.

For advertising practitioners, our finding highlighted the importance of flow experience and suggested every effort should be made to create personalized ads in an engaging, interactive, or creative manner to capture and maintain consumer attention, understanding, and enjoyment. Specifically, our study suggests informative messages can lead to a flow experience and a positive attitude. Therefore, providing customers with a message that they might regard as useful is important. To do so, cutting-edge solutions and systems for personalization should be applied, and more precise calculation is needed to send advertising message according to targeted customers' needs or interests. What is more, consumers' privacy concern at present is an important issue that should be treated with appropriate consideration. Advertising practitioners should have a clear understanding of privacy expectations in the Chinese context. Online advertising platform providers such as Weibo, WeChat, and DouYin should provide users with more control over privacy matters. For government organizations, effective ways for informing consumers about facts of online advertising and data tracking practices should be developed, and supervision of data collecting and usage process should be emphasized to generally reduce unnecessary privacy concerns.

## 6. Limitations and Future Research

The present study has several limitations that should be acknowledged. First, the sample population was restricted to China. Though it provides some insights into this rapidly developing market, more research is needed to generalize the research results to other cultural and country contexts. Second, this research utilized self-administered questionnaires to test the model which does not establish causality. Experimental methods should be adopted to further test the hypotheses posed by this study for future studies. Third, as personalized advertisements are tailored based on consumers' online behavior, and thus are likely to meet their needs, this study highlights informativeness in the advertising value model. Future studies can focus on the influence of other dimensions, such as entertainment and credibility. Finally, the dependent variable of this research is consumers' attitudes. Other behavior-related variables, such as consumers' intention to click on the ad, willingness to purchase the advertised products, and intention to share the ads should be considered in future research.

**Author Contributions:** Conceptualization, L.M. and Z.Y.; methodology, L.M.; software, L.M.; validation, X.Z. and Y.L.; formal analysis, X.Z.; investigation, X.Z. and Y.L.; data curation, Z.Y.; writing—original draft preparation, L.M.; writing—review and editing, Z.P.; visualization, Z.P. supervision, Z.Y.; funding acquisition, L.M. and Z.Y. All authors have read and agreed to the published version of the manuscript.

**Funding:** This research was funded by China Scholarship Council grant number 202008350025.

**Institutional Review Board Statement:** Not applicable.

**Informed Consent Statement:** Informed consent was obtained from all subjects involved in the study.

**Data Availability Statement:** Data can be accessed: https://figshare.com/s/877f36740c9f98aa8f50.

**Conflicts of Interest:** The authors declare no conflict of interest.

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
