# Peer review of "Consumers’ Attitudes towards Online Advertising: A Model of Personalization, Informativeness, Privacy Concern and Flow Experience"

_sustainability, doi:10.3390/su15054090_

Round 1

Reviewer 1 Report

This interesting paper reviews ways by which Consumers’ attitudes towards online advertising: a model of personalization, Informativeness, privacy concern and flow experience

Here are my suggestions:

1. The abstract lacks coherence and the method is not figured properly. Would suggest using shorter sentences.

2. I believe a wider description of privacy concern and flow experience is required in the introduction. Though you discuss examples in section 2+3, I think the reader needs to have more understanding of this matter at the start.

3. The method description is ambiguous and not well defined: The method is not figured properly- as this is a review, the method should be extremely specified. Some questions that have not been answered are, e.g.: Why did you use just one search engine, and why Google Scholar? How would you know/ estimate that you haven't missed important papers? How and why did you choose the criteria list? Why did you mention the "time-consuming process" and are you referring to the first reading or the second detailed reading?  college students are becoming an important 229 ? I would also refer to the goals of the search (section 3.3) once more in relation to the method, and explain why this method is most appropriate to achieve the goal.

4. Are sections 2.1+2.2.+2.3  all based on the reviewed literature? If so- it should be cleared, if not- section 3 and 4 seem like an introduction and maybe should appear before the method.

5. Not sure about "iResearch 2023 " being a valid source. I would suggest using a dictionary for definitions.

6. Did you look into the year of publication? As technologies change frequently this might be important.

7. Section 2.1 ( Online advertising and Perceived Personalization)  would specify the title. This seems like you are about to discuss the limitations of the study.

8: 4.3  line one is not cleared.

9. The manuscript is missing many sources. If statements are insights or assumptions of the author, they should be further explained and well established. The contribution of the author should be clear. For example, The mechanism of how perceived personalization affect marketing related outcomes is still in need of further exploring" Is it an assumption from the researcher or is it a gap? 

10. The language has issues grammatically as well regarding the requirement of a proper academic writing style.

Author Response

Dear reviewer:

Reviewer 2 Report

1. Revisit the hypothesis - most of the mediation hypotheses are conceptually wrong or misleading.

2. Hypothesis development must be adequate before you state the hypothesis for the study.

3. Draw the framework properly. You can use SPSS-AMOS drawing pad.

4. Show the measurement model (diagram) in SPSS-AMOS graphic for every constructs. Show the CFA results such as Fitness Indexes, Factor Loading and using the CFA results you assess for Construct Validity, Convergent Validity, Discriminant Validity and Composite Reliability. Show the process.

5. Draw the structural model in the SPSS-AMOS syntax. From here you input data and RUN. Show the output on the diagram. Using the output of structural model you explain the results to the audience.

6. Conduct hypothesis testing properly - show how you test direct effect, mediation effect etc. For mediation testing, you need to confirm the results using Bootstrapping procedure. Show the output in SPSS-AMOS Graphic. As of now, nothing from SPSS-AMOS as you claim to be using.

7. Do not mention AMOS - Need to mention IBM-SPSS-AMOS. And do not mixed-up between method and tools. You are modelling and analyzing your study using SEM (structural equation modelling) and the tool you employ is IBM-SPSS-AMOS. So, you need to report - the method employed for this study is structural equation modelling (SEM) in IBM-SPSS-AMOS version 28.0 (for example).

Author Response

Dear reviewer:

Reviewer 3 Report

The paper have certain research value for practical problems. However, The demonstration process of this paper is not rigorous enough and the research depth is insufficient, it does not meet the conditions for publication.

1. What is the significance of the study? The introductory section of your manuscript needs to highlight the significance of the study’s findings. Who are the intended beneficiaries of the study’s findings and how would they benefit from it?

2. In the last part of the introduction, I will advice that the differences from other studies should be comprehensively presented, and needs to emphasize this papers originality.

3. In the literature review, consider adding introduction to relevant theories and the recent literature.

4. Page 4. The figures name needs to be supplemented and the content in the figure should be clear and complete.

5. In 3.1 Questionnaire design of page 4, the content description is too simple, I advice that the authors can add survey object and sampling method.  

6. In 3.2 Measure of page 4. I think it is convenient to use charts to show the measurement content.

7. In 3.3 Data collection of page 5. Data collection process of the questionnaire should be scientific and critical content should not be removed at will. How many questionnaires are distributed? How many questionnaires are collected? How many valid questionnaires?

8. Who filled the questionnaires? What is the inclusion and exclusion criteria for the respondents to be qualified to participate in the study? 

9.  In 4.1, what is the conclusion of descriptive statistics? As a part of result, the authors should explicitly present research conclusions.

10. The respondents’ demographics has not been explained in the manuscript. This should be reported and discussed. 

11. Since the authors used five-point range scales throughout the questionnaire, can they provide explanations on how they addressed common method bias and sampling bias prior to data collection? Also, after data collection, how did the authors ensure that common method bias and sampling bias are within acceptable limits in the study using statistical methods?

12. In 4.3. When discussing the moderating relationship, try to refer to the graph. Discussions of the moderating effect should be done with the aid of the graph. This does not seem to be the case with this manuscript. Correct this accordingly.

13.  In this discussion section, it is important that when you start by stating if a relationship is significant or not, you should also go further by stating if it supports the hypothesis or not. 

14. When presenting practical implications, try to add appropriate suggestion from the customers perspective.

Author Response

Dear reviewer:

Round 2

Reviewer 1 Report

If Constructs were justified with better clarity in the earlier sections, then the discussion could have included clearer examples from the samples to support the coding categories/constructs. There needs to be a stronger case of responding to specific research question/s and its relationship to the theoretical framework.

Author Response

Dear Reviewer:

Reviewer 2 Report

I am still not convinced with the analysis. The Cronbach Alpha should be presented alongside CR and AVE since it is not the element of CFA. This is misleading. Furthermore, the measurement model in SPSS-AMOS is not presented. No output from the measurement model also. The factor loading, fitness indexes should be referred to the figure (measurement model). Show from where you obtained the figures in discussion. 

Also no structural model being presented. Should present how you convert framework into structural model, input data and run. Show the standardized regression and unstandardized regression output. From these output you discuss the results of R-square, testing hypothesis etc. 

The authors are presenting and discussing a lot of results but failed to refer  these results on the graphical output from Spss-Amos Graphics. Due to this weaknesses, I am still not convinced. Thus, the authors should do it again. Read my comment on the article also.

Author Response

Dear reviewer:

Reviewer 3 Report

The paper can be accepted.

Author Response

Dear Reviewer,

Thank you very much for your review.

Round 3

Reviewer 2 Report

I support this article to be published.

Author Response

Dear Reviewer:

Thank you very much for your valuable comments.